# Giant thermal expansion and α-precipitation pathways in Ti-alloys

Matthias Bönisch [1,2,7], Ajit Panigrahi[3,4], Mihai Stoica[1,8], Mariana Calin[1], Eike Ahrens[1], Michael Zehetbauer[3], Werner Skrotzki[2] & Jürgen Eckert[5,6]

Ti-alloys represent the principal structural materials in both aerospace development and metallic biomaterials. Key to optimizing their mechanical and functional behaviour is in-depth know-how of their phases and the complex interplay of diffusive vs. displacive phase transformations to permit the tailoring of intricate microstructures across a wide spectrum of configurations. Here, we report on structural changes and phase transformations of Ti–Nb alloys during heating by in situ synchrotron diffraction. These materials exhibit anisotropic thermal expansion yielding some of the largest linear expansion coefficients ($+163.9 \times 10^{-6}$ to $-95.1 \times 10^{-6}\ {}^\circ\mathrm{C}^{-1}$) ever reported. Moreover, we describe two pathways leading to the precipitation of the α-phase mediated by diffusion-based orthorhombic structures, $\alpha''_{lean}$ and $\alpha''_{iso}$. Via coupling the lattice parameters to composition both phases evolve into α through rejection of Nb. These findings have the potential to promote new microstructural design approaches for Ti–Nb alloys and β-stabilized Ti-alloys in general.

[1] IFW Dresden, Institute for Complex Materials, Helmholtzstraße 20, D-01069 Dresden, Germany. [2] Institute of Structural Physics, Technische Universität Dresden, Haeckelstraße 3, D-01062 Dresden, Germany. [3] Physics of Nanostructured Materials, University of Vienna, Boltzmanngasse 5, A-1090 Vienna, Austria. [4] Institute of Minerals and Materials Technology, Bhubaneswar 751013, India. [5] Erich Schmid Institute of Materials Science, Austrian Academy of Sciences (ÖAW), Jahnstraße 12, A-8700 Leoben, Austria. [6] Department Materials Physics, Montanuniversität Leoben, Jahnstraße 12, A-8700 Leoben, Austria. [7] Present address: Department of Mechanical Science and Engineering, University of Illinois at Urbana-Champaign, Urbana IL 61801, USA. [8] Present address: Laboratory of Metal Physics and Technology, Department of Materials, ETH Zürich, CH-8093 Zürich, Switzerland. Correspondence and requests for materials should be addressed to M.B. (email: mbonisch@illinois.edu)

Ti-alloys are the workhorses in modern aerospace design and engineering of metallic biomaterials. In particular the class of β-stabilized Ti-alloys represents highly promising multifunctional materials with desirable structural and functional properties for various biomedical and engineering applications[1–3]. Certain alloy compositions display low Young's moduli ($E$) below 80 GPa after rapid cooling[4, 5], providing suitable starting points for the development of novel low-modulus alloys for load-bearing implant applications. Due to their high strength to density ratio β-stabilized Ti-alloys have also become increasingly used for various aerospace applications such as airframes and landing gears[3].

Due to the thermoelastic martensitic transformation of the body centred cubic (bcc) β-phase to orthorhombic martensite α″ this alloy family also demonstrates shape memory (SM) behaviour and superelasticity (SE)[6, 7]. Up to the present day, these features have been stirring ever increasing interest.

The mechanical and functional properties of these alloys are optimized by controlled adjustment of the microstructural parameters via complex thermomechanical processing paths. For instance, the mechanical behaviour of β and near-β Ti alloys can be significantly improved by uniform dispersions of fine α and/or $\omega_{iso}$ precipitates[6, 8–10]. The transformation and precipitation pathways occurring during aging decide the morphology, size and arrangement of the precipitating products[11, 12].

The great diversity and complex interplay of diffusion-driven vs. displacive phase transformations in β-stabilized Ti-alloys stimulates their further exploration. For instance, several years ago a novel coupled diffusional-displacive transformation mechanism occurring during formation of isothermal $\omega_{iso}$ was described[13]. Recently, a unique nanolaminate structure consisting of α″ martensite and planar complexions of a thermal $\omega_{ath}$ was reported[14]. Furthermore, for two commercial β-stabilized Ti–Mo–Fe–Al and Ti–Al–Mo–V–Fe–Cr alloys, Ivasishin et al. observed the formation of an intermediate orthorhombic phase exhibiting the same crystal structure as α″ martensite during early stages of α-precipitation[10, 15]. More recently, employing in situ diffraction methods this phase was also detected upon heat treatment of Ti–Al–Mo–Cr–Sn–Zr, Ti–Al–Mo–V–Cr–(Zr) and Ti–V–Fe–Al alloys used for advanced structural aircraft components[11, 12, 16–18]. For some alloys a β-stabilizer lean structure, denoted α″$_{lean}$, was observed during the decomposition of α″ martensite[11, 19]. It was reported that decomposition of α″

martensite into α and β phases by aging below the austenite start temperature $A_s$ involves a spinodal mechanism[20, 21] implying continuous changes in chemical composition.

Detailed knowledge about transformation and precipitation processes is the foundation for the advancement of tailored thermomechanical treatment routes. However, following these processes through diffraction patterns recorded ex situ at room temperature (RT) is challenging. First, due to the variations in cell parameters caused by thermal expansion and secondly, because of transformations potentially taking place during cooling from the aging temperature to RT. Only when the phases resulting from aging neither decompose nor transform during cooling the microstructure formed at the aging temperature will be observable at RT. For the β-phase this only applies, when it contains sufficient β-stabilizers to suppress the martensite start temperature $M_s$ below RT. Furthermore, β needs to be stable enough against $\omega_{ath}$ formation and additional α precipitation. Otherwise precipitation of α and/or $\omega_{iso}$ as well as martensite formation takes place upon cooling after aging, thus modifying the constitution of phases at the temperature of interest[22]. These difficulties can be overcome by the use of in situ diffraction methods permitting direct observation of the microstructural evolution caused by heat treatments. In situ diffraction methods, employing, e.g. synchrotron radiation, are therefore the first choice to capture phase reactions and transitions taking place over narrow temperature and time intervals directly at the critical temperature[11, 12, 16–18, 23].

Since the discovery of SM in Ti–Nb[7] this system serves as a prototype to study SM and SE in Ni-free Ti-based alloys. Its importance is underlined by the fact that most of the recently developed low-modulus as well as SM and SE β-stabilized Ti-based alloys are modifications of the Ti–Nb system[6, 9, 23]. Hence, a better understanding of the binary base alloy system will help explain, at least to some extent, the alloys derived from it.

The transformation pathways and precipitation sequences triggered by heating of α″ martensite depend on the Nb content. This is illustrated in Fig. 1a which shows differential-scanning calorimetry (DSC) curves for four binary Ti–$c$Nb alloys ($c = 16$, 21, 28.5, 36 wt.%) adapted from previous studies[22, 24]. At the heating rate employed (10 °C min⁻¹), reversion of α″ martensite followed by substantial $\omega_{iso}$ precipitation occurs for $c \geq 28.5$ (Fig. 1a). In contrast, for $c \leq 21$ α″ decomposes directly into an α + β phase mixture. Formation of $\omega_{iso}$ starts during the martensitic

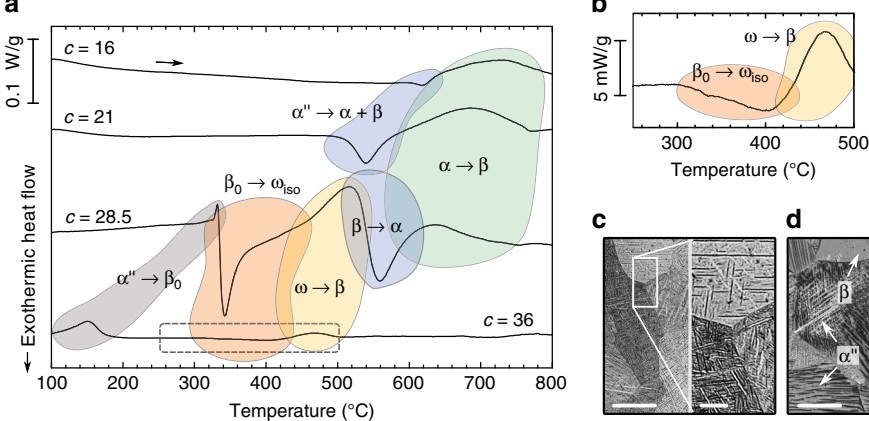

**Fig. 1** Overview of heating-induced transformation behaviour. **a** Isochronal heat flow curves recorded during heating of martensitic Ti-(16, 21, 28.5, 36)Nb alloys at 10 °C min⁻¹. **b** Close-up view of Ti-36Nb in the interval enclosed by the dashed lines in (**a**). Differential-scanning calorimetry curves were adapted from previous studies[22, 24]. Light microscopy images of homogenized and quenched **c** Ti-28.5Nb and **d** Ti-36Nb illustrate the fully α″-martensitic microstructure for $c = 16$, 21, 28.5 and the partially austenitic microstructure for $c = 36$. The scale bars of the low magnification images in (**c**) and (**d**) are 200 μm and that of the magnified portion in (**c**) is 50 μm

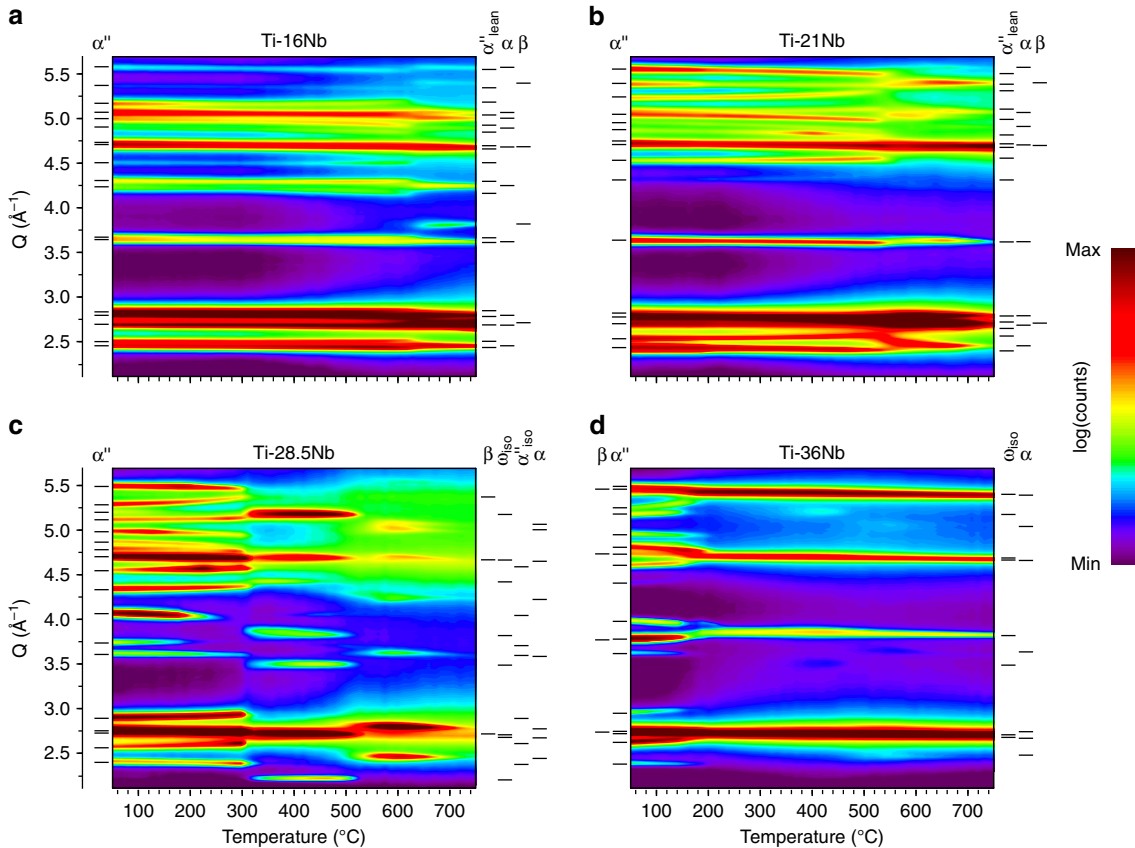

**Fig. 2** Evolution of in situ synchrotron X-ray diffraction patterns. Results for **a** Ti–16Nb, **b** Ti–21Nb, **c** Ti–28.5Nb and **d** Ti–36Nb during heating at 10 °C min$^{-1}$. On the left side of each panel the reflections of the phases present in the initial state at room temperature are identified. On the right side of each panel those formed during heating are identified. For each phase the markers indicate the reflection positions at their first appearance (i.e. for the lowest temperature)

reversion of $\alpha''$ for $c = 28.5$ whereas more than 100 °C above the austenite finish temperature $A_f$ for $c = 36$, as shown in Fig. 1b. During further heating $\omega_{iso}$ transforms back to $\beta$. For $c = 28.5$ this reaction overlaps and is followed by the precipitation of $\alpha$. After martensite decomposition and $\alpha$ precipitation, the $\alpha$-phase gradually transforms into $\beta$ upon approaching the $\alpha$–$\beta$ transus.

In the current study, we employ variable-temperature synchrotron X-ray diffraction (SXRD) to track these transformations in situ for the same alloy formulations and heating rate as in Fig. 1. The in situ data allow clearly demarcating (within the detection limit of SXRD) critical temperatures, such as start and end temperatures of concurrent reactions, where DSC otherwise provides only rough values. Furthermore, temperature-induced changes in the unit cells are easily revealed. In the first part of the paper we use the temperature dependence of the lattice parameters to analyze the thermal expansion of $\alpha''$ martensite in the Ti–Nb system. The data reveal a strong anisotropy and giant linear expansion of $|\alpha_L| \gtrsim 100 \times 10^{-6}\,°C^{-1}$ along certain crystallographic directions for the Nb richest alloy. Afterwards we examine the decomposition of $\alpha''$ martensite into $\alpha$- and $\beta$-phases through the formation of Nb-depleted $\alpha''_{lean}$. In the last part we investigate the orthorhombic precipitation product $\alpha''_{iso}$ and show for the first time that this structure also occurs in the Ti–Nb system prior to $\alpha$ formation during aging of metastable $\beta$.

## Results

**In situ observation of structural evolution during heating.** The initial microstructure consisted of orthorhombic martensite $\alpha''$

for $c = 16$, 21 and 28.5 and of $\alpha''$ with some austenitic $\beta$-phase for $c = 36$, in agreement with the literature[25, 26]. Figure 1c, d show corresponding light microscopy images. The martensite plates exhibited a thickness of up to 15 μm and lengths of up to several 100 μm, in some cases cutting across entire $\beta$ grains. The present alloys exhibit yield strengths and Young's moduli between 400–660 MPa and 65– 83 GPa, respectively, and can be plastically deformed to more than 20%[4, 5, 23].

Figure 2 presents the evolution of the X-ray diffractograms during heating to 760 °C, where several new phases appeared including $\alpha$, $\beta$, $\omega_{iso}$. In addition, Nb-depleted $\alpha''$, denoted $\alpha''_{lean}$, and a thermally formed phase exhibiting the same crystal structure as $\alpha''$, denoted $\alpha''_{iso}$, were observed. Using the diffractograms (Fig. 2) the lattice parameters were refined by Le Bail fits and the refinement results are presented in Fig. 3. For each composition Fig. 3 shows the evolution of the lattice parameters for all phases detected as well as the aspect ratios $b_{\alpha''}/a_{\alpha''}$ and $c_{\alpha''}/a_{\alpha''}$ for the three types of $\alpha''$ (martensite, $\alpha''_{lean}$, $\alpha''_{iso}$). The aspect ratios influence the shape of the orthorhombic $\alpha''$ unit cell and determine whether it resembles more closely $\alpha'$ or $\beta$. If $b_{\alpha''}/a_{\alpha''} = \sqrt{3} \cong 1.732$ the atoms on $(001)_{\alpha''}$ are arranged in a hexagonal pattern identical to the atoms on $(0001)_{\alpha'}$ for hexagonal close-packed (hcp) martensite $\alpha'$. On the other hand, for $b_{\alpha''}/a_{\alpha''} = c_{\alpha''}/a_{\alpha''} = \sqrt{2} \cong 1.414$ the crystal structure of $\alpha''$ is similar to bcc $\beta$[25, 26]. Both limiting values are indicated in Fig. 3. Crystal structures identical to hcp $\alpha'$ and of bcc $\beta$ are only obtained by additionally adjusting the position of atoms on $(002)_{\alpha''}$ through the fractional coordinate $y$ in Wyckoff position 4c of space group Cmcm[25].

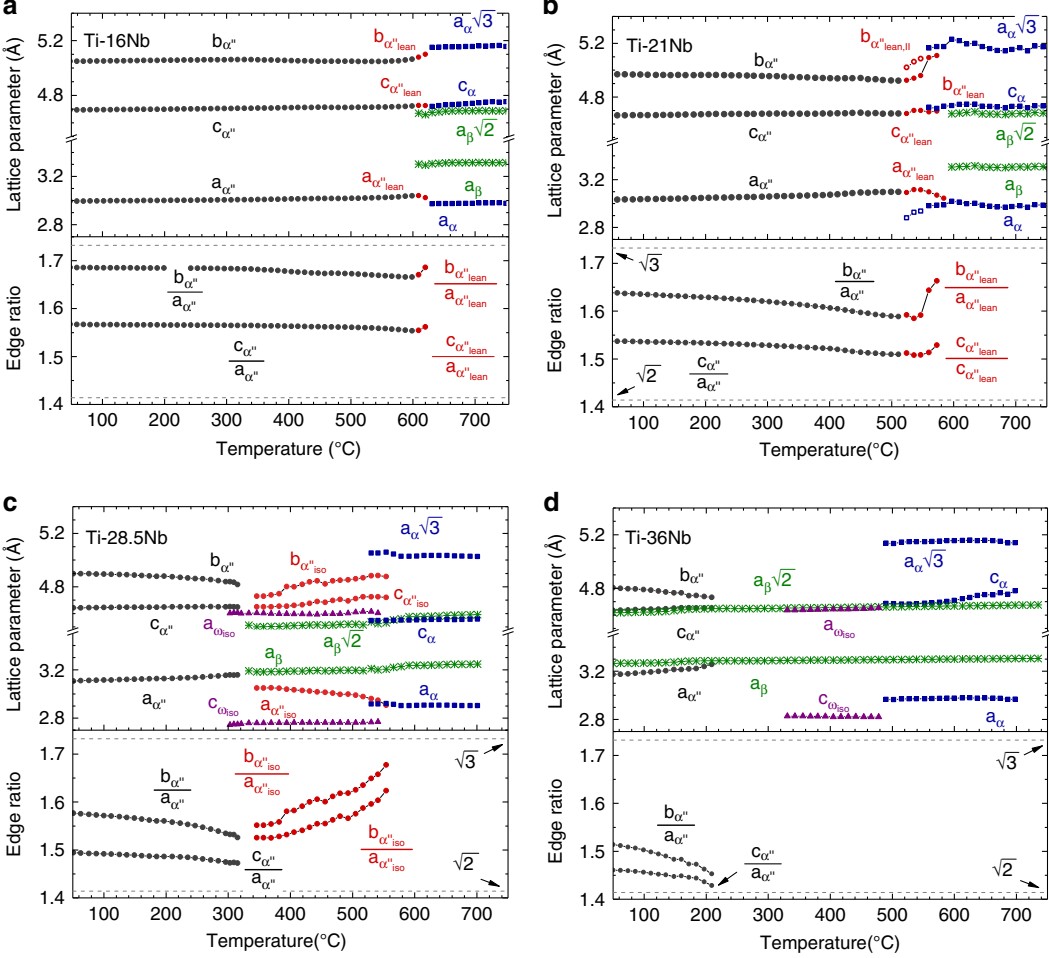

**Fig. 3** Heating-induced variation of unit cell geometries. Structural evolution during heating at 10 °C min$^{-1}$ for **a** Ti–16Nb **b** Ti–21Nb **c** Ti–28.5Nb and **d** Ti–36Nb. For each composition the upper panel shows the refined lattice parameters of all phases detected and the lower panel the ratios of $b$ and $c$ relative to $a$ for the orthorhombic phases. The refinement uncertainty is smaller than the markers

**Thermal Expansion of α″ Martensite**. The lattice parameters of α″ martensite are strongly affected by the Nb content. Likewise the shape of the orthorhombic unit cell of α″ varies with Nb content, as seen from $b_{\alpha''}/a_{\alpha''}$ and $c_{\alpha''}/a_{\alpha''}$ at 50 °C in Fig. 3: When the Nb content increases from $c = 16$ to $c = 36$, $b_{\alpha''}/a_{\alpha''}$ and $c_{\alpha''}/a_{\alpha''}$, respectively, drop from 1.686 to 1.514 and from 1.567 to 1.460, thus coming closer to the limit of 1.414 corresponding to β. Thus, Nb-lean α″ is structurally closer to hcp α′ whereas Nb-rich α″ is more similar to bcc β.

Increasing the Nb content reduces the temperature, below which α″ martensite present in the initial microstructure is preserved during heating: it decreases from 600 °C for $c = 16$ to 210 °C for $c = 36$ (Fig. 2 and Table 1). Figures 2 and 3 reveal that the reflections of α″ martensite and correspondingly its lattice parameters shift with temperature due to thermal expansion, the shift being stronger the more Nb is present. Furthermore, thermal expansion of α″ martensite is highly anisotropic and depends on the composition: The $a_{\alpha''}$ and $c_{\alpha''}$ spacings expand for all compositions while the $b_{\alpha''}$ spacing slightly expands for $c = 16$ but contracts otherwise. Table 1 lists the corresponding values of the linear thermal expansion coefficient $\alpha_L$ along the unit cell edges of α″ martensite for the present alloys as well as their volumetric thermal expansion coefficient $\alpha_V$. These values are compared with literature data[27–32], as discussed below. For Ti–36Nb the expansion coefficients of austenite β between 50 °C and 305 °C (no precipitation reactions were detected in this

interval, Figs. 1 and 2d) are reported as well. For each material in Table 1 (except for Ti–35.4Nb), the values given are average rates of expansion between $T_{low}$ and $T_{high}$ according to

$$\alpha = \frac{1}{X_{T_{low}}} \cdot \frac{X_{T_{high}} - X_{T_{low}}}{T_{high} - T_{low}} \qquad (1)$$

where X represents either a lattice parameter or the unit cell volume. In case of Ti–35.4 Nb and Co$_{49}$Ni$_{21}$Ga$_{30}$[30], the straight line functions for $\alpha_L$ were averaged across $T_{low}$–$T_{high}$. For Ni$_{54}$Mn$_{25}$Ga$_{21}$[31] $\alpha_L$ was calculated on the basis of the lattice parameters pertaining to $T_{low}$ and $T_{high}$.

For α″ martensite in the present alloys $\alpha_L$ along $a_{\alpha''}$ is larger than along $c_{\alpha''}$ and in both directions positive for all compositions. Contrary to this, for $c = (21, 28.5, 36)$ $\alpha_L$ along $b_{\alpha''}$ is negative, corresponding to a contraction of the $b_{\alpha''}$ spacing upon heating. The magnitude of $\alpha_L$ along $a_{\alpha''}$ and along $b_{\alpha''}$ grows with increasing Nb content. $\alpha_L$ along $c_{\alpha''}$ decreases up to $c = 28.5$, but is largest for $c = 36$.

This leads to a remarkable anisotropy of the thermal expansion of α″ martensite for Ti–36Nb: While the $a_{\alpha''}$ and $c_{\alpha''}$ spacings expand at a rate of 163.9×10$^{-6}$ °C$^{-1}$ and 24.4×10$^{-6}$ °C$^{-1}$, respectively, the $b_{\alpha''}$ spacing contracts by −95.1×10$^{-6}$ °C$^{-1}$ between 50 °C and 210 °C. These rates are comparable to or even larger in magnitude than those of materials exhibiting

**Table 1 Thermal expansion rates of martensitic Ti–Nb alloys compared with the literature**

| Material | Phase, crystal system | Temperature $T_{low}-T_{high}$ (°C) | $\alpha_L$ | | | $\alpha_V$ $10^{-6}$ °C$^{-1}$ | $b_{\alpha''}/a_{\alpha''}$ $10^{-6}$ °C$^{-1}$ | $c_{\alpha''}/a_{\alpha''}$ $10^{-6}$ °C$^{-1}$ | Ref. |
|---|---|---|---|---|---|---|---|---|---|
| | | | [100] $10^{-6}$ °C$^{-1}$ | [010] $10^{-6}$ °C$^{-1}$ | [001] $10^{-6}$ °C$^{-1}$ | | | | |
| Ti-16Nb | $\alpha''$, O | 50-600 | 25.7 | 4.7 | 10.6 | 41.3 | −20.7 | −14.9 | Present work |
| Ti-21Nb | $\alpha''$, O | 50-510 | 47.2 | −20.9 | 6.9 | 32.9 | −66.6 | −39.5 | Present work |
| Ti-28.5 Nb | $\alpha''$, O | 50-315 | 81.4 | −60.6 | 5.1 | 24.7 | −139.0 | −74.7 | Present work |
| Ti-36Nb | $\alpha''$, O | 50-210 | 163.9 | −95.1 | 24.4 | 91.0 | −252.3 | −135.9 | Present work |
| Ti-36Nb | β, C | 50-305 | 27.9 | 27.9 | 27.9 | 84.3 | | | Present work |
| Ti-35.4 Nb | $\alpha''$, O | 30-202 | 124.9 | −93.1 | 24.3 | 56.1 | | | 30 |
| Ti$_{50.5}$Ni$_{19.5}$Pd$_{30}$ | B19, O | 30-160 | 51.3 | −3.2 | −34.5 | 13.6 | | | 30 |
| Ti$_{50.1}$Ni$_{49.9}$ | B19', M | 25-100 | −47.2 | 43.8 | 22.7 | | | | 32 |
| Ni$_{54}$Mn$_{25}$Ga$_{21}$ | T | 52-218 | 85.1 | 85.1 | −105.6 | 62.8 | | | 31 |
| Co$_{49}$Ni$_{21}$Ga$_{30}$ | L1$_0$, T | 4 K-285 K | −42.5 | −42.5 | 60.0 | −25.1 | | | 30 |
| Ag$_3$[Co(CN)$_6$] | R | 20 K-500 K | ~136 | ~136 | ~−128 | | | | 29 |
| Sm$_{2.75}$C$_{60}$ | O | 4 K-32 K | ~−100 | ~−100 | ~−99 | ~−300 | | | 28 |
| Xe (solid) | C | 50 K-75 K | 235 | 235 | 235 | ~710 | | | 27 |

Linear and volumetric thermal expansion coefficients of $\alpha''$ martensite for Ti-(16-36)Nb compared with those of the martensitic B19 and B19' phases in TiNi-based alloys[30, 32], tetragonal martensite in Heusler (Co,Mn)GaNi[30, 31] alloys and with materials exhibiting some of the largest isotropic (positive and negative) and strongest anisotropic thermal expansion in crystalline solids known[27-29]. The expansion coefficients of austenite β for Ti-36Nb are reported as well. For each material the crystal system is indicated in the second column: O orthorhombic, M monoclinic, R rhombohedral, C cubic, T tetragonal. All values are means across the temperature ranges indicated in the third column and those of the present work exhibit relative uncertainties <10%. For the present alloys, the rates of the relative change of $b_{\alpha''}/a_{\alpha''}$ and $c_{\alpha''}/a_{\alpha''}$ with temperature are given in the last two columns

colossal thermal expansion (defined as $|\alpha_L| \geq 100 \times 10^{-6}$ °C$^{-1}$) such as Ag$_3$[Co(CN)$_6$][29, 33]. The present values for the expansion rates of the $b_{\alpha''}$ and $c_{\alpha''}$ spacings fully agree with recently published data for $\alpha''$ martensite in Ti–35.4Nb[30] (Table 1). The expansion rate along $c_{\alpha''}$ is somewhat lower[30] than in the present case which may be caused by fitting of individual reflections to determine the thermal expansion in contrast to the full pattern approach employed in the present work.

The TiNi-based martensites B19 and B19'[30, 32], which are closely related to $\alpha''$, and, as recently revealed, tetragonal martensite in Heusler (Co,Mn)NiGa alloys[30, 31] exhibit similarly strong anisotropic thermal expansion. Still, in most cases their expansion rates are smaller in magnitude than for Ti–36Nb and Ti–28.5Nb (Table 1). Such large shifts of martensite lattice parameter towards the parent phase in the vicinity of the transformation may be rationalized by taking a dynamical view of the martensite lattice paying attention to its phonon free energy[34]. Stresses originating from self-accommodation that are released by approaching the transition temperature may as well play part in the observed shifts.

The thermal expansion rates of $\alpha''$ (particularly for Ti–36Nb) are among the largest (both positive and negative) ever reported for solid crystalline metallic systems, be they isotropic or anisotropic. Typical values for $\alpha_L$ for engineering metals and alloys are positive and range between $0-40 \times 10^{-6}$ °C$^{-1}$[35]. Expansion rates comparable to or larger than those for Ti–36Nb are only found for members of other material classes but not for metallic systems. Representative examples are the ceramic framework material Ag$_3$[Co(CN)$_6$][29], the fulleride Sm$_{2.75}$C$_{60}$[28] and the extremely loosely bound solid Xe below 75 K[27] (Table 1).

The variation of the unit cell volume due to anisotropic thermal expansion may be positive or negative[36]. In the present case of $\alpha''$ martensite in Ti–Nb, the expansion and contraction along the unit cell edges partially compensate each other. Yet, the positive $\alpha_L$ along $a_{\alpha''}$ and $c_{\alpha''}$ dominate over the negative $\alpha_L$ along $b_{\alpha''}$ leading to an expansion rate $\alpha_V$ of the bulk between $24.7 \times 10^{-6}$ °C$^{-1}$ and $91.0 \times 10^{-6}$ °C$^{-1}$. The bulk expansion of β-phase in Ti–36Nb turns out slightly smaller ($84.3 \times 10^{-6}$ °C$^{-1}$) than that of $\alpha''$ ($91.0 \times 10^{-6}$ °C$^{-1}$).

**Martensite decomposition.** Depending on composition the disappearance of $\alpha''$ martensite during heating involves different transformation sequences and the formation of different phases. For $c = (16, 21)$ decomposition of $\alpha''$ into α and β phases takes place, which is accompanied by an exothermic event (Fig. 1). The high temperature diffraction data (Fig. 2) allow following the process of $\alpha''$ decomposition into α and β phases observed for Ti-(16,21)Nb in situ. Figure 4 presents close-ups of the diffraction patterns for the temperature intervals in which decomposition occurred. The refined lattice parameters for each temperature are shown in Fig. 3a, b. In the decomposition interval $\alpha''$ is denoted with the subscript lean, as justified below.

$\alpha''$ martensite is preserved in Ti–16Nb up to 600 °C and in Ti–21Nb up to 510 °C. No precipitation products were detected below these temperatures suggesting that $\alpha''$ maintains its initial composition present at RT. $\alpha''$ fully decomposes into α and β by further heating of Ti–16Nb to 630 °C and of Ti–21Nb to 590 °C. During decomposition, the lattice parameters of $\alpha''_{lean}$ change rapidly and the variation of the unit cell aspect ratios ($b_{\alpha''}/a_{\alpha''}$ and $c_{\alpha''}/a_{\alpha''}$) with temperature is inverted relative to the thermal expansion behaviour discussed previously. While upon heating below the temperature interval of decomposition $\alpha''$ martensite becomes more similar to austenite β, it becomes more α-like during decomposition. This is reflected by the behaviour of $b_{\alpha''}/a_{\alpha''}$ and $c_{\alpha''}/a_{\alpha''}$ which first decrease towards $\sqrt{2}$ with increasing temperature but then grow towards $\sqrt{3}$ during decomposition.

Comparison of the lattice parameters of $\alpha''_{lean}$ with those of $\alpha''$ martensite at RT suggests that $\alpha''_{lean}$ progressively rejects Nb thereby approaching equilibrium α. In this process $a_{\alpha''_{lean}}$ evolves into $a_\alpha$, $b_{\alpha''_{lean}}$ evolves into $a_\alpha\sqrt{3}$ and $c_{\alpha''_{lean}}$ develops into $c_\alpha$ (Fig. 3a, b). The occurrence of martensite decomposition is contingent on a sufficiently long dwell below $A_s$ or, correspondingly, on a sufficiently slow rate of heating as indicated by a study on Ti–10V–2Fe–3Al[11]: While heating at 20 °C min$^{-1}$ and below led to martensite decomposition and $\alpha''_{lean}$ formation, heating at 50 °C min$^{-1}$ triggered martensite reversion.

For a large fraction of the martensitic matrix of Ti–21Nb Nb depletion proceeds continuously as evidenced by the smooth changes of the $\alpha''$ lattice parameters towards those of α and of

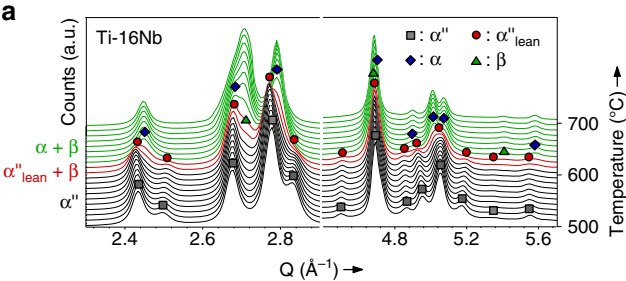

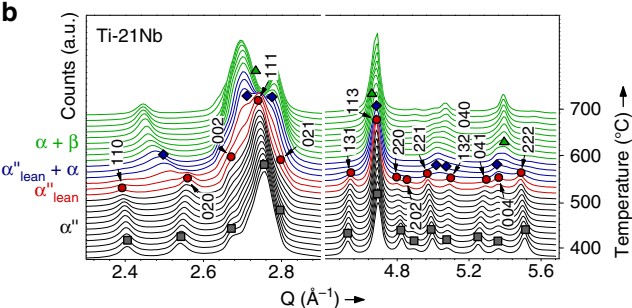

**Fig. 4** Martensite decomposition. In situ synchrotron X-ray diffraction patterns during isochronal heating (10 °C min⁻¹) showing the decomposition of α″ martensite into α and β phases for **a** Ti–16Nb and **b** Ti–21Nb. For Ti–21Nb the reflections of α″$_{lean}$ are exemplarily indexed

$(b_{\alpha''_{lean}})/(a_{\alpha''_{lean}})$ towards $\sqrt{3}$ (Fig. 3b). The smoothly changing lattice parameters of α″$_{lean}$ during decomposition agree with the notion of a spinodal decomposition mechanism of α″, which has been previously observed in Ti–V, Ti–Mo and Ti–Nb alloys upon isothermal aging below $A_s$[20, 21]. Yet, it needs to be pointed out that these smooth changes may as well indirectly stem from nucleation of a solute enriched phase, such as Nb-rich α″$_{rich}$ or β. Very small volume fractions of these phases will lead to only minor variations of the α″$_{lean}$ lattice parameters, which may give rise to continuous shifts of Bragg reflections due to the finite angle resolution of the diffractometer.

A close look at the diffraction patterns for Ti–21Nb in Fig. 4b indeed evidences that nucleation of α from α″ occurs in addition to a potential spinodal mechanism. As shown in Fig. 5a, upon heating of α″ above 520 °C the diffracted intensity at $Q \cong 2.52$ Å⁻¹ increases, which does not match any Bragg angle of α″. This reflection grows in intensity, shifts to lower $Q$ upon heating and gradually develops into $\{10\bar{1}0\}_\alpha$. Above this temperature the diffraction patterns clearly exhibit the hexagonal symmetry of hcp α (plus bcc β) and no contributions of α″ are required to fully model the observed intensities. The initial evolution of the $\{10\bar{1}0\}_\alpha$ reflection below 560 °C (indicated by the black arrow in Fig. 5a) may correspond to $\{10\bar{1}0\}_\alpha$ of an early form of α as well as to $(020)_{\alpha''}$ of a second orthorhombic component α″$_{lean,II}$. Since for both phases the modeled profiles reached comparable agreement with the experimental data it is problematic to unambiguously assign one of these phases.

Assigning the intensity at $Q \cong 2.52$ Å⁻¹ below 560 °C (black arrow in Fig. 5a) to the $\{10\bar{1}0\}_\alpha$ family of planes gives values for $a_\alpha$ which are smaller than $a_{\alpha''}$ by up to 7% (Fig. 5b). They are indicated by open blue squares in Figs. 3b and 5b. This rather large difference between $a_\alpha$ and $a_{\alpha''}$ implies that α formed from α″ by a regular nucleation process.

The equally valid assignment of $(020)_{\alpha''}$ to the intensity at $Q \cong 2.52$ Å⁻¹ in Fig. 5a infers the nucleation of a second orthorhombic Nb-lean component, denoted α″$_{lean,II}$. The corresponding values of $b_{\alpha''_{lean,II}}$ are larger than $b_{\alpha''_{lean}}$ by up to 2.5% (Fig. 5b). They are indicated as open red circles in Figs. 3b and 5b. Because it is

depleted of Nb the crystal structure of the second orthorhombic component is already closer to hcp α than that of its orthorhombic matrix. The crystal structures of both orthorhombic components, α″$_{lean}$ and α″$_{lean,II}$, continuously approach that of hcp α until at about 590 °C all α″$_{lean}$ components have transformed into α.

**Formation of α″$_{iso}$.** In contrast to the decomposition of α″ martensite into α and β phases during heating for $c = 16$ and 21, α″ reverts martensitically to austenite β$_0$ causing an endothermic event for $c = 28.5$ and 36 (Fig. 1a). For $c = 28.5$ martensite reversion is accompanied by the immediate formation of ω$_{iso}$ resulting in a ω$_{iso}$ + β phase mixture, as shown by the in situ diffractograms in Fig. 6. Aside from ω$_{iso}$ and α, they reveal the formation of an additional precipitation product, indicated α″$_{iso}$ and resembling α″, when heating continues after martensite reversion.

Above 340 °C the diffracted intensity increases at several angles agreeing with the crystal structure of α″ (five of these are indicated by arrows in Fig. 6). We call this precipitated phase α″$_{iso}$, as it exhibits the same crystal structure as α″ martensite but, in contrast to martensite, forms and evolves by a diffusion-based demixing process. We chose the symbol α″$_{iso}$, where the subscript *iso* stands for isothermal, purely for the purpose of consistency with previous studies in the literature where this phase has been observed. It must be emphasized that isothermal conditions[10, 17, 18, 37] do not constitute a prerequisite for α″$_{iso}$ formation[11, 12, 15, 16].

In Ti–36Nb α″$_{iso}$ was not encountered, which can be rationalized by the lower diffusivity of Nb for Ti–36Nb compared to Ti–28.5 Nb[26]. The reduced atomic mobility for $c = 36$ also shows itself in the delayed and weak precipitation of ω$_{iso}$ for this composition: the heat release associated with ω$_{iso}$ formation for $c = 36$ is many times smaller than for $c = 28.5$ and peaks at 400 °C for $c = 36$ whereas at 240 °C for $c = 28.5$, as shown in Fig. 1.

The refined lattice parameters and unit cell aspect ratios of α″$_{iso}$ for Ti–28.5Nb are shown in Fig. 3c. They exhibit a strong variation with temperature and display the same behaviour upon heating as the lattice parameters of α″$_{lean}$ for Ti–16Nb and Ti–21Nb: $a_{\alpha''_{iso}}$ shrinks, while all other parameters ($b_{\alpha''_{iso}}$, $c_{\alpha''_{iso}}$ as well as $b_{\alpha''_{iso}}/a_{\alpha''_{iso}}$ and $c_{\alpha''_{iso}}/a_{\alpha''_{iso}}$) grow and $b_{\alpha''_{iso}}/a_{\alpha''_{iso}}$ and $c_{\alpha''_{iso}}/a_{\alpha''_{iso}}$ approach $\sqrt{3}$ when the temperature increases. This implies that the crystal structure of α″$_{iso}$ progressively becomes more similar to hcp α. A comparable evolution of α″$_{iso}$ towards α when exposed to elevated temperatures was reported for several other β-stabilized Ti-alloys[10–12, 15, 16, 37]. In some cases the phase fraction of α″$_{iso}$ reaches more than 50 wt.% as reported for Ti–5Al–4Mo–4Cr–2Sn–2Zr (Ti17) under isothermal conditions[37].

The variation of the unit cell geometry of α″$_{iso}$ with temperature is inverted relative to the structural change of α″ martensite due to thermal expansion. This strongly suggest that compositional changes take place in α″$_{iso}$. For instance, for a Ti–Mo-based alloy (TIMETAL-LCB) analytical scanning transmission electron microscopy clearly evidenced that the β-stabilizer content of α″$_{iso}$ precipitates was progressively reduced with aging time[10].

The observed shift in the character of the α″$_{iso}$ unit cell towards a more hcp-like structure (Fig. 3c) indicates the expulsion of Nb atoms. Assuming that the cell parameters of α″$_{iso}$ change with the same rate as a function of Nb as those of α″ martensite[25] we estimated the change in composition of α″$_{iso}$ during continuous heating. Between 350 °C and 550 °C $a_{\alpha''_{iso}}$ shrinks by 0.14 Å, whereas $b_{\alpha''_{iso}}$ and $c_{\alpha''_{iso}}$ expand by 0.15 Å and 0.07 Å, respectively. These changes correspond to a reduction in the Nb content of

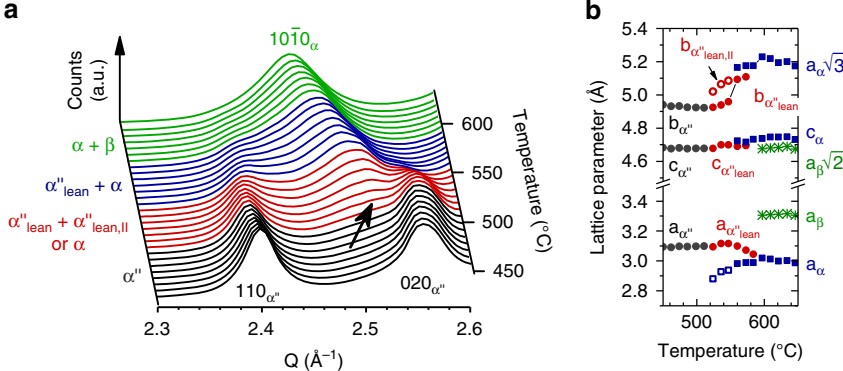

**Fig. 5** Close-up of the temperature interval of martensite decomposition for Ti–21Nb. **a** At 520 °C the diffracted intensity at $Q \cong 2.52$ Å$^{-1}$ increases corresponding to either (020) of a second orthorhombic component or to $\{10\bar{1}0\}_\alpha$. **b** The lattice parameters of the nucleating $\alpha''_{lean, II}$ and $\alpha$ are indicated by open symbols

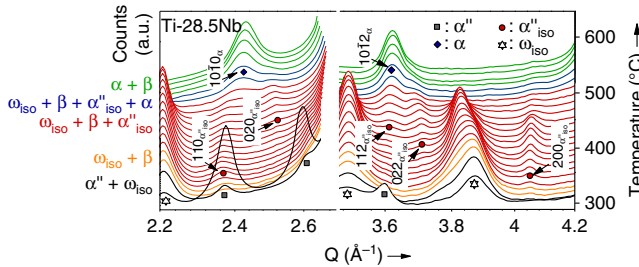

**Fig. 6** Formation of $\alpha''_{iso}$. In situ synchrotron X-ray diffraction patterns of Ti–28.5 Nb heated at 10 °C min$^{-1}$ showing the formation of $\alpha''_{iso}$ above martensite reversion and its gradual evolution into $\alpha$ during further heating

$\alpha''_{iso}$ by 12 to 23 wt.% and necessarily require long-range diffusion causing $\alpha''_{iso}$ to evolve into $\alpha$. Figure 6 clearly demonstrates this evolution where the $\{110\}_{\alpha''_{iso}}$ and $(020)_{\alpha''_{iso}}$ reflections and the $\{112\}_{\alpha''_{iso}}$ and $\{022\}_{\alpha''_{iso}}$ reflections merge into $\{10\bar{1}0\}_\alpha$ and $\{10\bar{1}2\}_\alpha$, respectively.

## Discussion

Aside from reaching giant values in case of Nb-rich alloy formulations, the thermal expansion rates of the present alloys can be adjusted across a wide range by simply modifying their composition. This represents an advantage over many ceramic materials which often show rather limited controllability of their thermal expansion[38]. Moreover, materials with negative thermal expansion (linear and/or volumetric) are of particular interest, since they permit designing composites with zero net thermal expansion when combined with positive thermal expansion materials.

The temperature dependence of the lattice parameters of $\alpha''$ martensite reveal further that its anisotropic thermal expansion causes $\alpha''$ to become more similar to austenite $\beta$ upon heating. A comparable behaviour was recently observed in several other martensitic systems[11, 30]. For each of the present alloys this process takes place across the entire temperature range in which $\alpha''$ martensite is present. Irrespective of composition, $b_{\alpha''}/a_{\alpha''}$ and $c_{\alpha''}/a_{\alpha''}$ continuously decrease with temperature towards 1.414 (Fig. 3) and the higher the Nb content the larger is the rate of change (Table 1). Thus, increasing the temperature exerts the same influence on the $\alpha''$ structure as adding Nb. In both cases, austenite $\beta$ becomes stabilized relative to $\alpha''$ martensite and at the same time the structural difference between $\alpha''$ and $\beta$ shrinks.

The polycrystalline nature of the irradiated volume (as illustrated in Supplementary Fig. 1) did not allow extraction of reliable information about the atom positions on the $(002)_{\alpha''}$ plane depending on the temperature. Nevertheless, it seems reasonable to assume that at high temperatures atoms on $(002)_{\alpha''}$ take positions which further increase the similarity of $\alpha''$ to $\beta$. This would manifest itself in an increase of the fractional coordinate $y$ of Wyckoff position 4c of space group Cmcm when $\alpha''$ is heated towards $A_f$[25].

The in situ data clearly evidences that during martensite decomposition $\alpha''_{lean}$ forms. The data indicate two decomposition sequences for Ti–21Nb, slightly different from each other:

(1) $\alpha'' \rightarrow \alpha''_{lean} + \alpha \ (+\alpha''_{rich} + \beta) \rightarrow \alpha + \beta$
(2) $\alpha'' \rightarrow \alpha''_{lean} + \alpha''_{lean,II} \ (+\alpha''_{rich} + \beta) \rightarrow \alpha + \beta$

For both pathways Nb-depleted $\alpha''_{lean}$ as well as Nb-enriched $\alpha''_{rich}$ form. Nb-enriched $\alpha''_{rich}$ and/or $\beta$ is expected to take up the Nb expelled from $\alpha''_{lean}$. $\alpha''_{rich}$ eventually evolves into $\beta$. Although $\alpha''_{rich}$ and $\beta$ were not detected for Ti–21Nb, the presence of at least one of these phases is necessary for mass conservation during Nb-expulsion from $\alpha''_{lean}$. In addition, in the first pathway $\alpha$ nucleates, whereas in the second pathway a second Nb-lean component $\alpha''_{lean,II}$ nucleates and develops into $\alpha$.

The decomposition process of $\alpha''$ for $c = 16$ is very similar to that for $c = 21$. Again Nb-depleted $\alpha''_{lean}$ forms from the quenched-in $\alpha''$. However, no evidence for $\alpha$ or $\alpha''_{lean,II}$ was found and the transformation sequence observed for Ti–16Nb during heating was thus:

$$\alpha'' \rightarrow \alpha''_{lean} + \beta \rightarrow \alpha + \beta$$

When heating continues after $\alpha''$ decomposition the reflections of $\alpha$ gradually weaken, which corresponds to the endothermic conversion of $\alpha$ to $\beta$ while approaching the $\alpha$–$\beta$ transus (Fig. 1).

In case of Ti–28.5Nb, precipitation of $\alpha$ proceeds through the diffusion-based formation of $\alpha''_{iso}$. Comparing the lattice parameters of $\alpha''_{iso}$ (Fig. 3c) with those measured for $\alpha''$ martensite at RT[25] reveals a slight difference in the geometry of their orthorhombic unit cells. For a given aspect ratio $b_{\alpha''_{iso}}/a_{\alpha''_{iso}} = b_{\alpha''}/a_{\alpha''}$ of the orthorhombic unit cell, $c_{\alpha''_{iso}}/a_{\alpha''_{iso}}$ of the diffusion-mediated $\alpha''_{iso}$ is larger by 3 to 4 % than the corresponding $c_{\alpha''}/a_{\alpha''}$ of $\alpha''$ martensite. In other words, the difference between $b$ and $c$ is smaller for $\alpha''_{iso}$ than for $\alpha''$ and, thus, the character of the $\alpha''_{iso}$ unit cell is more tetragonal than that of the $\alpha''$ unit cell.

One may speculate that the underlying reason is the constraining nature of the $\beta$-matrix which does not allow $\alpha''_{iso}$ to adjust its unit cell shape freely. A recent study[39] of $\alpha''_{iso}$ formation in Ti-5553 supports this assumption. Using phase-field

**Table 2 Space group and Wyckoff position(s) for each phase used for the refinements**

| Phase | Space group (No.) | Occupied Wyckoff position(s) |
|---|---|---|
| α | P6$_3$/mmc (194) | 2a |
| α″ martensite, α″$_{lean}$, α″$_{iso}$ | Cmcm (63) | 4c |
| β | Im$\bar{3}$m (229) | 2a |
| ω$_{iso}$ | P6/mmm (191) | 1a, 2d |

simulations the authors concluded that internal stresses lead to the observed cell parameters of α″$_{iso}$. Allowing for self-accommodation of differently oriented α″$_{iso}$ variants did not alter their conclusion.

We have thus demonstrated that α″ martensite displays both one of the largest positive and one of the largest negative thermal expansion coefficients reported for any metallic material to date.

Furthermore, we observed for the first time that at elevated temperatures prior to α formation, rejection of Nb from the parent phases leads to the diffusion-based formation of the intermediate structures α″$_{lean}$ and α″$_{iso}$. Both phases continuously evolve into α as they expel Nb into their surrounding matrices. Controlled activation of these diffusion-based processes will allow tailoring of novel microstructures. These discoveries will have far-reaching consequences by expanding the application range of β-stabilized Ti-alloys and by opening up new opportunities to improve their mechanical and functional behaviour.

## Methods

**Alloy synthesis**. Four binary Ti–cNb alloys with Nb contents of c = 16, 21, 28.5 and 36 wt.% were chosen for this study and cast into rods (10 mm diameter) by arc-melting followed by cold-crucible casting[22]. Analysis of the Nb and O contents of the as-cast alloys by wet chemical analysis and hot gas extraction showed that the deviation from the nominal Nb content and the maximal O content was <0.1 wt.% each. The cast rods were homogenized for 24 h at 1000 °C under Ar and quenched into water (HQ treatment). Prior to quenching the parent β-phase exhibited a grain size of (300 ± 150) μm. Specimens for light microscopy were prepared by mechanical polishing followed by etching in an aqueous solution of 2 vol.% HF and 6.5 vol.% HNO$_3$.

**In situ synchrotron X-ray diffraction**. In situ SXRD was conducted at the ID11 beamline of the European Synchrotron Radiation Facility (ESRF) in Grenoble, France. Thin rod-shaped samples (diameter ≈ 800 μm) were extracted from the HQ material by electrical discharge machining and subjected to a solution treatment at 1000 °C for 4 h in Ar followed by water quenching. In situ heating of the samples from RT to 760 °C at a rate of 10 °C min$^{-1}$ was achieved with a resistively heated modified Linkam hot stage purged with Ar. Diffraction patterns were recorded by a 2-dimensional (2D) image-plate detector (FReLoN) with 2048 × 2048 pixels centered on the transmitted beam. Example raw 2D-diffraction patterns are shown in Supplementary Fig. 1. An X-ray wavelength λ = 0.20664 Å and a beam cross-section of 50 × 50 μm$^2$ were used. The sample-to-detector distance and zero-point shift were calibrated at RT using standard CeO$_2$ powder. Data collection was done every °C using an exposure time of 0.2 s. Calibration, background subtraction and azimuthal integration of the 2D patterns into 1-dimensional patterns were performed with Fit2D[40]. Azimuthal integration was carried out over the entire 360° in order to minimize potential effects of texture and graininess. Diffraction angles are reported in terms of the wavevector transfer Q = 4π•sin(θ)/λ, where θ denotes the semi-angle between the incident and the diffracted beam.

**Analysis of diffraction patterns**. On the patterns obtained phase analysis was carried out. For each phase the space group and occupied Wyckoff position(s) are listed in Table 2. The same orthorhombic space group (Cmcm) was used for the 3 orthorhombic α″-like structures (α″ martensite, α″$_{lean}$ and α″$_{iso}$). For each phase detected the lattice parameters were determined depending on temperature by structureless (Le Bail) refinements utilizing the FULLPROF programme suite[41]. The reflection profiles were simulated with Thompson–Cox–Hastings pseudo-Voigt functions[42] and the background was modelled by linear segments, which were included in the refinements.

**Data availability**. The datasets generated and analyzed during the current study are available from the corresponding author on request.

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

## Acknowledgements

We acknowledge the use of the ID11 beamline at the ESRF for in situ X-ray diffraction and local support by J. Wright and N. Harker. We thank E. Schafler and T. Waitz for helpful discussions as well as S. Neumann and H.-P. Trinks for technical support. This work received funding from the European Commission within the framework of FP7/2007–13 grant agreement No. 264635 (BioTiNet-ITN) and from the Graduate Academy of the TU Dresden within the Excellence Initiative of the federal and state governments of Germany. Additional support through the German Science Foundation (DFG) under the Leibniz Program (grant EC 111/26-1), the European Research Council under the ERC Advanced Grant INTELHYB (grant ERC-2013-ADG-340025) is gratefully acknowledged.

## Author contributions

M.B., M.S., M.C. and J.E. designed this research. M.B., A.P. and M.S. performed the experiments, M.B., A.P., M.S. and E.A. analyzed the data and M.B. wrote the manuscript. All authors discussed the results and M.C., M.Z., W.S. and J.E. supervised the study.
