## [Peer Review File · Nature Communications]

Reviewers' Comments:

Reviewer #1:

Remarks to the Author:

The draft entitled 'Colossal thermal expansion and alpha-precipitation pathways in Ti-alloys' by M. Bönisch and co-workers is a careful experimental investigation of the transformations of beta Ti-Nb alloys by means of in-situ high energy Xray diffraction (mainly).

The two major results of the paper are:

1/ the so-called colossal thermal expansion of the orthorhombic martensite alpha';

2/ a comparison of the transformation pathways during heating of different Ti-Nb alloys.

These claims are original and supported by a deep analysis of the experimental results and a thorough discussion.

From my point of view, the high quality of the paper is sufficient to be considered for publication, with only few minor revisions.

1/ I understand that it is necessary to claim some exceptional properties to have a better chance to be published in prestigious journals. But I am not sure that it is so fair to claim that the Ti-Nb alloys (in particular with 36%Nb) display colossal thermal expansions. Of course, two of the coefficients measured along the three directions of the orthorhombic lattice are really huge, but the effective thermal expansion drops below $100 \times 10^{-6} / ^\circ\text{C}$ (and around 40 for all alloys except Ti-36%Nb), compared to the fulleride $\text{Sm}_{2.76}\text{C}_{60}$ and Xe. Moreover, I wonder whether this so-called colossal thermal expansion can be observed in macroscopic specimen, if the six variants of the alpha' phase are present. I understand that the authors comply with the previous works that have used such emphatic claims, but I feel uncomfortable with this trend of overstatement.

2/ I have found the discussion quite confusing on pages 10-11 about what happens to alpha''_lean: spinodal decomposition or nucleation. In particular, at the very beginning of the concerned paragraph, it is written that 'the smoothly changing lattice parameters of alpha''_lean during decomposition indicate a likewise smooth change in composition, which is a fingerprint of a spinodal mechanism'. I find this claim quite misleading, because the smooth changes are monitored only on average lattice parameters and can be induced by discontinuous events (in terms of concentration) starting in small proportion that increases progressively.

I suggest to clarify the whole paragraph by stating clearly what are the observations in favor of spinodal decomposition and those in favor to nucleation.

3/ Finally, I suggest to add one figure with an enlargement of Fig. 3b, showing only the lattice parameters (not the ratios) in the temperature range where two different indexations are proposed (closed and open symbols).

Reviewer #2:

Remarks to the Author:

This manuscript reported the structural changes and phase transformations of Ti-Nb alloys during heating by in-situ synchrotron diffraction. It has been demonstrated these alloys exhibit exceptional anisotropic thermal expansion yielding some of the largest linear expansion coefficients, and two unique transformation pathways leading to the precipitation of alpha-phase mediated by diffusion-based orthorhombic structures, alpha''_lean and alpha''_iso. Via the coupling of the lattice parameters to composition both phases continuously evolve into alpha through rejection of the beta-stabilizer. The paper is well organized and clearly presented. The new findings are original, sound, and of interest. These new findings have potential to promote novel applications and microstructural design approaches not only for Ti-Nb alloys but for beta-stabilized Ti-alloys in general.

Reviewer #3:

Remarks to the Author:

This is interesting, and I should thank the authors for pointing me to the Ivasishin 2005 paper, which I wasn't aware of. Their analysis of that paper is quite insightful.

The main claim of the paper is (1) that orthorhombic alpha double prime martensite has very large changes in lattice parameter, over quite large temperature ranges, on heating prior to transformation to the austenite phase.

Claim (2) is that the decomposition can involve a diffusional component in some of the Ti-Nb alloys examined; as two phases are formed this is uncontroversial and has been inferred before by the authors referenced.

Claim (3) is that the transformation pathway is novel, which I don't think it really is, but its nice work nevertheless. I don't really see the justification for separating the alpha' lean from the decomposing alpha, but this is fine.

Claim (1) is not completely new - the authors give some examples but similarly large thermal expansions/contractions along particular crystal axes relaxing towards the parent austenite have also been seen in other SMAs including ZrCuCoNi and Ni₂MnGa (author: Azeem MA). In fact, if one takes a dynamical view of the theory of martensites (MT Dove, Structure and Dynamics: An atomic view of Materials, OUP 2003) then it would seem unsurprising both that the modulus of a martensite and its lattice parameter might shift rapidly towards the parent phase in the vicinity of transformation. Phonon-type considerations would take one along the same line of argument (e.g. Petry's work on Ti, Zr and on NiTi). As the authors say, the accommodation stresses with the matrix may also play a part, as the twin-accommodated laths dissolve.

In that sense, connecting to phase field modelling and plotting the volume fraction evolution would be helpful.

What would be exciting would be a well argued theoretical rationale, along the lines sketched out above or otherwise, for WHY this observation seems to be commonly made in martensites and why only some of them show this effect.

So, I think its fine and basically correct.

The topic area is slightly obscure and I confess I'm not sure the audience is that large. Because the volume thermal expansion is quite large, it is still possible to argue that this is one of the largest thermal expansions, albeit anisotropic, seen in any material, and over quite a useful temperature range. If it survived repeated thermal cycling (seems unlikely), it might even be of practical utility, although this is quite an expensive alloy.

One can see in the Ivasishin paper that there has been a line of thought of using these transformations as nucleation sites to make very strong, tough and ductile titanium alloys like LCB, and this is probably the future.

In addition, there has been a lot of interest in this particular martensite over the last decade, following the Saito, Niinomi etc line of work, mostly in making low modulus biomedical alloys where it wasn't initially welcomed! So there is a substantial community interested in these transformation pathways in titanium.

In that sense its probably a good paper for this journal - papers in this topic area do seem to get cited.

I also wonder if having some microscopy and some baseline characterisation - grain sizes, mechanical data etc - wouldn't help the reader, rather than just sXRD. It would also be nice to see

the whole area detector data in order to satisfy oneself about texture effects, orientation dependence, grain size etc.

In order to boost the impact of the paper, I think my main recommendation to the authors would be to see if they can't shorten the text on pp10-15, which is rather involved and a bit heavy.

Final point: the key Figure is Figure 3 - and its rather too squished to read. Same for Fig 4a. I really had to zoom in on screen to see the a'lean - which i think is just a continuation of a'. Authors need to fix for readability. I'd like to see the colour scales in Fig 2 sorted out so the reader can see the minor phases more easily. Supplementary info should show some raw 2D detector images.

Detailed Responses to Reviewers' Comments by Author

[REVIEWER'S COMMENT 1]:

Reviewer #1 (Remarks to the Author):

The draft entitled ``Colossal thermal expansion and alpha-precipitation pathways in Ti-alloys' by M. Bönisch and co-workers is a careful experimental investigation of the transformations of beta Ti-Nb alloys by means of in-situ high energy Xray diffraction (mainly).

The two major results of the paper are:

- 1/ the so-called colossal thermal expansion of the orthorhombic martensite alpha';
- 2/ a comparison of the transformation pathways during heating of different Ti-Nb alloys.

These claims are original and supported by a deep analysis of the experimental results and a thorough discussion.

From my point of view, the high quality of the paper is sufficient to be considered for publication, with only few minor revisions.

1/ I understand that it is necessary to claim some exceptional properties to have a better chance to be published in prestigious journals. But I am not sure that it is so fair to claim that the Ti-Nb alloys (in particular with 36%Nb) display colossal thermal expansions. Of course, two of the coefficients measured along the three directions of the orthorhombic lattice are really huge, but the effective thermal expansion drops below $100 \times 10^{-6} / ^\circ\text{C}$ (and around 40 for all alloys except Ti-36%Nb), compared to the fulleride Sm_{2.76}C₆₀ and Xe.

Moreover, I wonder whether this so-called colossal thermal expansion can be observed in macroscopic specimen, if the six variants of the alpha' phase are present. I understand that the authors comply with the previous works that have used such emphatic claims, but I feel uncomfortable with this trend of overstatement.

[AUTHORS REPLY 1]:

We thank the reviewer for his constructive critique and positive recommendation. In what follows we address each of the comments and suggestions in detail.

The term 'colossal' was first used by Goodwin et al. in their 2008 study of the framework material Ag₃[Co(CN)₆] published in Science 319, 794. They used this term to highlight the extraordinary linear thermal expansion of trigonal Ag₃[Co(CN)₆] to signify crystallographic expansion rates $|\alpha_L| \geq 100 \times 10^{-6} \text{ K}^{-1}$. Their measurements revealed a positive expansion rate between $+130 \times 10^{-6} \text{ K}^{-1}$ and $+150 \times 10^{-6} \text{ K}^{-1}$ for the lattice parameter *a* and a negative rate between $-120 \times 10^{-6} \text{ K}^{-1}$ and $-130 \times 10^{-6} \text{ K}^{-1}$ for *c*. The pertaining volumetric expansion α_V ranges from $+130 \times 10^{-6} \text{ K}^{-1}$ to $+180 \times 10^{-6} \text{ K}^{-1}$ and, correspondingly, the effective linear expansion (i.e. for a polycrystal with random texture) given by $\alpha_V/3$ lies between $+60 \times 10^{-6} \text{ K}^{-1}$ and $+43 \times 10^{-6} \text{ K}^{-1}$.

In our present work, we adopted this definition, since for Ti-36Nb martensite α'' exhibits $|\alpha_L| \geq 100 \times 10^{-6} \text{ K}^{-1}$ for at least one crystallographic direction and an effective thermal linear expansion of $+30 \times 10^{-6} \text{ K}^{-1}$. At the same time, we understand and share the reviewer's concern about the trend of overstatement in the scientific literature. For this reason and because the remaining alloys studied exhibit large to very large expansion values, yet smaller than $100 \times 10^{-6} \text{ K}^{-1}$, we replaced the term colossal by giant.

In doing so the following edits were carried out:

Title: The title was modified into 'Giant thermal expansion and α -precipitation pathways in Ti-alloys'.

Line 109: The data reveal a strong anisotropy and giant linear expansion of $|\alpha_L| \gtrsim 100 \cdot 10^{-6} / ^\circ\text{C}$ along certain crystallographic directions for the Nb richest alloy.

Furthermore, the linear thermal expansion of a polycrystalline aggregate of a material exhibiting different expansion rates along different crystallographic directions depends on the aggregate's texture. In case of a random arrangement of the martensite variants in a macroscopic specimen the effective linear expansion will be approximately $\alpha\sqrt{3}$. For the present alloys $8 \cdot 10^{-6}/^{\circ}\text{C} < \alpha\sqrt{3} < 30 \cdot 10^{-6}/^{\circ}\text{C}$ (Table 1). Controlled adjustment of the texture may lead to observation of the giant expansion in macroscopic specimens.

[REVIEWER'S COMMENT 2]:

2/ I have found the discussion quite confusing on pages 10-11 about what happens to α''_{lean} : spinodal decomposition or nucleation. In particular, at the very beginning of the concerned paragraph, it is written that "the smoothly changing lattice parameters of α''_{lean} during decomposition indicate a likewise smooth change in composition, which is a fingerprint of a spinodal mechanism". I find this claim quite misleading, because the smooth changes are monitored only on average lattice parameters and can be induced by discontinuous events (in terms of concentration) starting in small proportion that increases progressively.

I suggest to clarify the whole paragraph by stating clearly what are the observations in favor of spinodal decomposition and those in favor to nucleation.

[AUTHORS REPLY 2]:

We shall thank the reviewer for his precise observation and for pointing out that smooth position changes of XRD reflections can also be caused by discontinuous events. This may happen e.g. in case of a secondary phase precipitating in small amounts that lead to minute changes of the lattice parameters or may be due to the finite resolution of the diffractometer unable to separate the reflections. We fully agree with this comment. To incorporate this aspect into the discussion and to clarify the discussion on pages 10-11 we modified and shortened the paragraphs on pages 10-11 as indicated below:

Line 244 onwards:

For a large fraction of the martensitic matrix of Ti-21Nb Nb depletion proceeds continuously as evidenced by the smooth changes of the α''_{lean} lattice parameters towards those of α and of $(b_{\alpha''_{lean}})/(a_{\alpha''_{lean}})$ towards $\sqrt{3}$ (Fig. 3b). The smoothly changing lattice parameters of α''_{lean} during decomposition agree with the notion of a spinodal decomposition mechanism of α'' , which has been previously observed in Ti-V, Ti-Mo and Ti-Nb alloys upon isothermal aging below $A_s^{20,21}$. Yet, it needs to be pointed out that these smooth changes may as well indirectly stem from nucleation of a solute enriched phase, such as Nb-rich α''_{rich} or β . Very small volume fractions of these phases will lead to only minute variations of the α''_{lean} lattice parameters, which may give rise to continuous shifts of Bragg reflections due to the finite angle resolution of the diffractometer.

A close look at the diffraction patterns for Ti-21Nb in Fig. 4b indeed evidences that nucleation of α from α'' occurs in addition to a potential spinodal mechanism. As shown in Fig. 5a, upon heating of α'' above 520°C the diffracted intensity at $Q \cong 2.52 \text{ \AA}^{-1}$ increases,... The initial evolution of the $\{10\bar{1}0\}_{\alpha}$ reflection below 560°C (indicated by the black arrow in Fig. 5a) may correspond to $\{10\bar{1}0\}_{\alpha}$ of an early form of α as well as to $(020)_{\alpha''}$ of a second orthorhombic component $\alpha''_{lean,II}$.

Line 265:

This rather large difference between a_{α} and $a_{\alpha''}$ implies that α formed from α'' by a regular nucleation process.

Line 267:

The equally valid assignment of $(020)_{\alpha''}$ to the intensity at $Q \cong 2.52 \text{ \AA}^{-1}$ in Fig. 5a infers the nucleation of a second orthorhombic Nb-lean component, denoted $\alpha''_{lean,II}$. The corresponding values

of $b_{\alpha''_{lean,II}}$ are larger than $b_{\alpha''_{lean}}$ by up to 2.5% (Fig. 5b). They are indicated as open red circles in Fig. 3b and Fig. 5b. ...all α''_{lean} components have transformed into α .

Line 351:

In addition, in the first pathway α nucleates, whereas in the second pathway a second Nb-lean component $\alpha''_{lean,II}$ nucleates and develops into α .

To make the discussion more succinct and straightforward (as also suggested by Reviewer 3) the following sentences were removed:

~~*In case of spinodal decomposition small composition fluctuations are progressively amplified creating solute rich and solute poor regions. The solute concentration smoothly varies from the enriched to the depleted regions. On the other hand, if the precipitating phase forms by nucleation the product will exhibit its (meta)stable composition already from the earliest stages on³⁵. Consequently, for α nucleation the entailing changes in composition and lattice parameters are expected to be abrupt and sharp across the transformation front. However, if nucleation involves relatively large composition changes and diffusion is slow, undercritical nuclei (called embryos or clusters) may be observed³⁶. Their structure and chemical composition will be intermediate between the parent and the product phase....Hence, the continuous variations of the lattice parameters for α''_{lean} may as well result from the evolution of undercritical nuclei into stable α nuclei.*~~

Former references 35 and 36 were deleted accordingly.

[REVIEWER'S COMMENT 3]:

3/ Finally, I suggest to add one figure with an enlargement of Fig. 3b, showing only the lattice parameters (not the ratios) in the temperature range where two different indexations are proposed (closed and open symbols).

[AUTHORS REPLY 3]:

Following the reviewer's suggestion we added an enlargement of the decomposition interval in Fig. 3b to more clearly show the lattice parameters. The new figure appears in the manuscript as Fig. 5b, next to the close-up of the corresponding X-ray diffractograms. To accommodate these changes, parts of the caption of Fig. 5 were modified:

Line 237:

At 520°C the diffracted intensity at $Q \cong 2.52 \text{ \AA}^{-1}$ increases corresponding to either (020) of a second orthorhombic component or to $\{10\bar{1}0\}_{\alpha}$. (b) The lattice parameters of the nucleating $\alpha''_{lean,II}$ and α are indicated by open symbols.

In order to provide a consistent presentation along with the enlargement of Fig. 3b, Fig. 5a (close-up of the X-ray diffractograms) and Fig. 4b were adapted. In these figures the initial part of the decomposition interval (in red) was changed from α''_{lean} to $\alpha''_{lean} + \alpha''_{lean,II}$ or α . Furthermore, to avoid potential confusion $\alpha''_{lean,I}$ was replaced by α''_{lean} throughout the manuscript (in total 2 appearances, Line 272 and Line 346).

[REVIEWER'S COMMENT 4]:

Reviewer #2 (Remarks to the Author):

This manuscript reported the structural changes and phase transformations of Ti-Nb alloys during heating by in-situ synchrotron diffraction. It has been demonstrated these alloys exhibit exceptional anisotropic thermal expansion yielding some of the largest linear expansion coefficients, and two unique transformation pathways leading to the precipitation of α -phase mediated by diffusion-based orthorhombic structures, α'' lean and α'' iso. Via the coupling of the lattice parameters to composition both phases continuously evolve into α through rejection of the β -stabilizer. The paper is well organized and clearly presented. The new findings are original, sound, and of interest. These new findings have potential to promote novel applications and microstructural design approaches not only for Ti-Nb alloys but for β -stabilized Ti-alloys in general.

[AUTHORS REPLY 4]:

We thank the reviewer very much for taking the time to examine our manuscript and highly appreciate the reviewer's positive reception of the manuscript.

[REVIEWER'S COMMENT 5]:

Reviewer #3 (Remarks to the Author):

This is interesting, and I should thank the authors for pointing me to the Ivasishin 2005 paper, which I wasn't aware of. Their analysis of that paper is quite insightful.

The main claim of the paper is (1) that orthorhombic alpha double prime martensite has very large changes in lattice parameter, over quite large temperature ranges, on heating prior to transformation to the austenite phase.

Claim (2) is that the decomposition can involve a diffusional component in some of the Ti-Nb alloys examined; as two phases are formed this is uncontroversial and has been inferred before by the authors referenced.

Claim (3) is that the transformation pathway is novel, which I don't think it really is, but its nice work nevertheless. I don't really see the justification for separating the alpha' lean from the decomposing alpha, but this is fine.

[AUTHORS REPLY 5]:

We would like thank the reviewer for his very careful and high quality review. Below we address each comment point-by-point.

By using the symbols α''_{lean} and α'' we aim to distinguish α'' martensite from the decomposition product α''_{lean} . α'' martensite exhibits the quenched-in composition of the high temperature β phase and forms by a purely athermal process from austenite β . α''_{lean} exhibits the same crystal structure (identical space group and populated Wyckoff positions) as martensite α'' . However, it has another composition than the quenched in α'' martensite due to the diffusion based solute depletion (Nb in the present case). In that sense it can be regarded as a modification of α'' but not as martensite, which forms per definition without diffusion. To signify that the composition of α''_{lean} differs from its parent structure (α'') due to diffusion and to differentiate between their formation mechanisms we use the subscript lean.

[REVIEWER'S COMMENT 6]:

Claim (1) is not completely new - the authors give some examples but similarly large thermal expansions/contractions along particular crystal axes relaxing towards the parent austenite have also been seen in other SMAs including ZrCuCoNi and Ni₂MnGa (author: Azeem MA). In fact, if one takes a dynamical view of the theory of martensites (MT Dove, Structure and Dynamics: An atomic view of Materials, OUP 2003) then it would seem unsurprising both that the modulus of a martensite and its lattice parameter might shift rapidly towards the parent phase in the vicinity of transformation. Phonon-type considerations would take one along the same line of argument (e.g. Petry's work on Ti, Zr and on NiTi). As the authors say, the accommodation stresses with the matrix may also play a part, as the twin-accommodated laths dissolve.

In that sense, connecting to phase field modelling and plotting the volume fraction evolution would be helpful.

What would be exciting would be a well argued theoretical rationale, along the lines sketched out above or otherwise, for WHY this observation seems to be commonly made in martensites and why only some of them show this effect.

So, I think its fine and basically correct.

[AUTHORS REPLY 6]:

It is truly intriguing that several martensites show remarkable thermal expansion properties, either in terms of anisotropy or magnitude, or both. We shall thank the reviewer for pointing out the recent studies by Azeem et al. which further testify this behaviour. We included their work on Ni₂MnGa and along the same line the data on CoNiGa by Monroe et al. in our reference list and the table of thermal expansion coefficients (Table 1).

In doing so the following adaptations were carried out:

Caption of Table 1 (Line 554):

Linear and volumetric thermal expansion coefficients of α'' martensite for Ti-(16-36)Nb compared with those of the martensitic B19 and B19' phases in TiNi-based alloys^{30,32}, tetragonal martensite in Heusler (Co,Mn)GaNi^{30,31} alloys....

Line 560:

T- tetragonal was added.

Table 1 contains thus two more rows presenting the expansion coefficients of Ni₅₄Mn₂₅Ga₂₁ and Co₄₉Ni₂₁Ga₃₀.

Line 173:

In case of Ti-35.4Nb and Co₄₉Ni₂₁Ga₃₀³⁰, the straight line functions for α_L were averaged across $T_{low} - T_{high}$. For Ni₅₄Mn₂₅Ga₂₁³¹ α_L was calculated on the basis of the lattice parameters pertaining to T_{low} and T_{high} .

Line 189:

The TiNi-based martensites B19 and B19^{30,32}, which are closely related to α'' , and, as recently revealed, tetragonal martensite in Heusler (Co,Mn)NiGa alloys^{30,31} exhibit similarly strong anisotropic thermal expansion.

We also thank the reviewer for indicating the chapter on the dynamical theory of displacive transitions in Martin T. Dove's book. It is a highly valuable read, which we do not want to withhold from the readers. We agree that, aside from the intrinsic order (or lattice) parameter changes, the observed shifts may also be affected the accommodation stresses, as mentioned by the reviewer. To expand these lines of thought, we inserted the following sentences:

Line 192:

Such large shifts of martensite lattice parameter towards the parent phase in the vicinity of the transformation may be rationalized by taking a dynamical view of the martensite lattice paying attention to its phonon free energy³⁴. Stresses originating from self-accommodation that are released by approaching the transition temperature may as well play part in the observed shifts.

We agree with and appreciate the suggestion to utilize phase field modelling to gain a better understanding of the unit cell changes and their origin (e.g. intrinsic factors vs. matrix constraints). To gain a better understanding of the thermal expansion of martensites in general (e.g. to find out why some martensites show strong lattice parameter shifts while others don't) will be an exciting line of research to follow. We believe incorporation of these aspects into the (already longer than average) manuscript will broaden its scope too much and weaken its overall clarity and conciseness. Therefore, we wish to leave these aspects for future studies allowing giving attention to these issues in the necessary detail.

[REVIEWER'S COMMENT 7]:

The topic area is slightly obscure and I confess I'm not sure the audience is that large. Because the volume thermal expansion is quite large, it is still possible to argue that this is one of the largest thermal expansions, albeit anisotropic, seen in any material, and over quite a useful temperature range. If it survived repeated thermal cycling (seems unlikely), it might even be of practical utility, although this is quite an expensive alloy.

[AUTHORS REPLY 7]:

The study of thermal expansion is a continuously growing field as documented by recent reviews (C. Lind, Materials 2012, 5, 1125-1154; Chen et al. Chem. Soc. Rev., 2015, 44, 3522). Materials with negative thermal expansion (linear and/or volumetric) are of particular interest, since they allow designing composites with zero thermal expansion when combined with positive thermal expansion materials. In this sense we believe that the presented results are of great relevance for the study of

thermal expansion and the development of materials with tailored temperature-induced dimensional changes.

Aside from exhibiting giant expansion rates in case of Nb-rich alloy formulations, the thermal expansion rates of the present alloys can be adjusted across a wide range by modifying their composition. This represents an advantage over many ceramic materials which often show rather limited controllability of their thermal expansion (Chen et al. Chem. Soc. Rev., 2015, 44, 3522). We agree, the Nb-rich alloys are relatively expensive due to their Nb content, yet they exhibit beneficial properties (e.g. high corrosion resistance, high formability) that may render them superior compared to cheaper alloys for application that require e.g. corrosion resistant, dimension critical components and/or thermally driven actuators. Low-cost alloys that display giant thermal expansion may be found by replacing Nb with cheaper beta-stabilizers, such as V, which show the same martensitic bcc to orthorhombic transformation.

We included the key aspects of the discussion above into the manuscript. In doing so one reference has been added (Chen et al. Chem. Soc. Rev., 2015, 44, 3522).

Line 321:

Aside from reaching giant values in case of Nb-rich alloy formulations, the thermal expansion rates of the present alloys can be adjusted across a wide range by simply modifying their composition. This represents an advantage over many ceramic materials which often show rather limited controllability of their thermal expansion³⁸. Moreover, materials with negative thermal expansion (linear and/or volumetric) are of particular interest, since they permit designing composites with zero net thermal expansion when combined with positive thermal expansion materials.
The...

[REVIEWER'S COMMENT 8]:

One can see in the Ivasishin paper that there has been a line of thought of using these transformations as nucleation sites to make very strong, tough and ductile titanium alloys like LCB, and this is probably the future.

In addition, there has been a lot of interest in this particular martensite over the last decade, following the Saito, Niinomi etc line of work, mostly in making low modulus biomedical alloys where it wasn't initially welcomed! So there is a substantial community interested in these transformation pathways in titanium.

In that sense its probably a good paper for this journal - papers in this topic area do seem to get cited.

[AUTHORS REPLY 8]:

We thank the reviewer for his constructive opinion. Our thoughts go into the same direction: using these transformations to either directly strengthen the beta phase or martensite, or to create nucleation sites for other phases (alpha, omega) to realize ductile high strength Ti-alloys respectively functional or low-modulus alloys. Phase transformations of beta-stabilized Ti-alloys are heavily researched these days and extensive endeavours are made to uncover transformation pathways and mechanisms, as testified by the growing number of publications being published on these topics over the last years.

[REVIEWER'S COMMENT 9]:

I also wonder if having some microscopy and some baseline characterisation - grain sizes, mechanical data etc - wouldn't help the reader, rather than just sXRD. It would also be nice to see the whole area detector data in order to satisfy oneself about texture effects, orientation dependence, grain size etc.

[AUTHORS REPLY 9]:

We thank the reviewer for his valuable suggestion. Since some readers may be interested in basic microstructural and mechanical properties of the alloys investigated, we included this information in the manuscript or make appropriate reference to the literature. In doing so, we inserted light microscopy images of the fully martensitic alloy Ti-28Nb (representative for Ti-16Nb, Ti-21Nb and Ti-28Nb) and of the martensitic + beta microstructure of Ti-36Nb into Figure 1. These images appear as subfigures c and d. The mean grain size of the parent beta phase prior to quenching is (300 +-

150) μm , the α'' martensite plates exhibit a thickness of up to 15 μm and lengths of up to several 100 μm , in some cases spanning entire parent beta grains. The four alloys investigated in this study have yield strengths of 400–660 MPa and Young's moduli of 65–83 GPa, while being deformable to more than 20% [Hanada, S. et al. Mater. Sci. Forum 426–432, 3103–3108 (2003); Bönisch et al. Mater. Sci. Eng. C 48, 511–520 (2015); Bönisch, M. Structural properties, deformation behavior and thermal stability of martensitic Ti-Nb alloys. Dissertation, Technische Universität Dresden (2016)]. Furthermore, we added a new figure (Supplementary Fig. 1) showing the entire detector area of several raw 2D diffraction patterns. For each alloy we selected 2 images, the first being always the starting condition at 50°C. At the temperature of the second images for Ti-16Nb (620°C) and for Ti-21Nb (570°C) martensite decomposition is in progress. The second image for Ti-16Nb (480°C) corresponds to α''_{iso} formation. The second image for Ti-36Nb shows α'' martensite at elevated temperature (185°C). These images illustrate the graininess and potential local texture of the irradiated volume. In order to minimize texture and graininess effects on the intensity in 1D-diffractograms the 2D-patterns were integrated over the entire azimuth of 360°.

By incorporating this information we modified the manuscript as follows:

Added to caption of Fig. 1:

DSC curves were adapted from previous studies^{22,24}. Light microscopy images of HO (c) Ti-28.5Nb and (d) Ti-36Nb illustrate the fully α'' -martensitic microstructure for $c = 16, 21, 28.5$ and the partially austenitic microstructure for $c = 36$. The scale bars of the low magnification images in (c) and (d) are 200 μm and that of the magnified portion in (c) is 50 μm .

Line 118 onward:

Figs. 1c and 1d show corresponding light microscopy images. The martensite plates exhibited a thickness of up to 15 μm and lengths of up to several 100 μm , in some cases cutting across entire β grains. The present alloys exhibit yield strengths and Young's moduli between 400–660 MPa and 65–83 GPa, respectively, and can be plastically deformed to more than 20%^{23,4,5}. Fig. 2 presents the evolution of the X-ray diffractograms during heating to 760°C, where several new phases appeared including α , β , ω_{iso} .

Lines 388:

Prior to quenching the parent β -phase exhibited a grain size of (300 \pm 150) μm . Specimens for light microscopy were prepared by mechanical polishing followed by etching in an aqueous solution of 2 vol.% HF and 6.5 vol.% HNO₃.

Line 336:

The polycrystalline nature of the irradiated volume (as illustrated in Supplementary Fig. 1)...

Line 398:

Example raw 2D-diffraction patterns are shown in Supplementary Fig. 1.

Line 402:

Azimuthal integration was carried out over the entire 360° in order to minimize potential effects of texture and graininess.

[REVIEWER'S COMMENT 10]:

In order to boost the impact of the paper, I think my main recommendation to the authors would be to see if they can't shorten the text on pp10-15, which is rather involved and a bit heavy.

[AUTHORS REPLY 10]:

We are very thankful for this comment. Following the reviewer's recommendation and to make the manuscript text more succinct, we shortened the text from section 'Martensite decomposition' onwards. In particular, the following sentences were removed and formulations modified aiming to improve the overall clarity.

~~*In case of spinodal decomposition small composition fluctuations are progressively amplified creating solute rich and solute poor regions. The solute concentration smoothly varies from the enriched to the*~~

~~depleted regions. On the other hand, if the precipitating phase forms by nucleation the product will exhibit its (meta)stable composition already from the earliest stages on³⁵. Consequently, for α nucleation the entailing changes in composition and lattice parameters are expected to be abrupt and sharp across the transformation front. However, if nucleation involves relatively large composition changes and diffusion is slow, undercritical nuclei (called embryos or clusters) may be observed³⁶. Their structure and chemical composition will be intermediate between the parent and the product phase.... Hence, the continuous variations of the lattice parameters for α''_{lean} may as well result from the evolution of undercritical nuclei into stable α nuclei.~~

~~...as variable temperature conditions, such as continuous heating like in the present and other works, give equally rise to its formation.~~

Line 287:

~~In Ti-36Nb, the alloy with the highest Nb content investigated in this work, α''_{iso} was not encountered, ...~~

Line 307:

~~The variation of the unit cell geometry of α''_{iso} is inverted relative to the structural change of α'' martensite due to thermal expansion strongly suggesting that compositional changes take place in α''_{iso} . For instance, for a Ti-Mo based alloy (TIMETAL-LCB) analytical scanning transmission~~

Line 316:

~~Combining each value with its corresponding rate of change with Nb content gives These changes correspond to a reduction in the Nb content of α''_{iso} by 12 to 23 wt.% in the aforementioned temperature interval. and necessarily require long-range diffusion causing α''_{iso} to evolve into α .~~

~~... and can be deduced from the temperature dependent lattice parameter data of α'' martensite for Ti-10V-2Fe-3Al.~~

Line 332:

~~Thus, increasing the temperature at constant e exerts the same influence on the α'' structure as adding Nb at constant temperature. In both cases, ... by coming closer to A_T ...~~

~~... thereby resembling the observed trend of γ on approaching A_T at constant temperature by raising the Nb content.~~

Line 342:

~~...during martensite decomposition of α'' β -stabilizer lean- α''_{lean} forms.~~

Line 348:

~~For both pathways Nb-depleted α''_{lean} as well as Nb-enriched α''_{rich} form. (designated α''_{lean} and α''_{rich} respectively). For both pathways, Nb-enriched...~~

Line 354:

~~However, no evidence for α or $\alpha''_{lean,II}$ was found and instead the β phase occurred already during the decomposition process. the transformation sequence observed for Ti-16Nb during heating was thus:~~

Line 359:

~~When heating of Ti-16Nb and Ti-21Nb continues ...~~

Line 365:

~~Conversely, for a given $c_{\alpha''_{iso}}/a_{\alpha''_{iso}} = c_{\alpha''}/a_{\alpha''}$, $b_{\alpha''_{iso}}/a_{\alpha''_{iso}}$ is smaller by 4 to 6 % than the corresponding $b_{\alpha''}/a_{\alpha''}$. In other words, the difference between the cell parameters b and c is smaller for α''_{iso} than for α'' . Thus, when compared with a hypothetical tetragonal unit cell with its edges~~

~~aligned parallel to the edges of the orthorhombic cell and its 4 fold axis oriented along $\frac{a''}{a''_{iso}} // a_{a-5}$ and, thus, the character of the α''_{iso} unit cell is more tetragonal than that of the α'' unit cell.~~

~~The arising internal stresses may hinder α''_{iso} to attain the unit cell geometry of α'' at a given Nb content.~~

Line 376:

~~...of the intermediate structures α''_{lean} and in case of martensite reversion to α''_{iso} .~~

[REVIEWER'S COMMENT 11]:

Final point: the key Figure is Figure 3 - and its rather too squished to read. Same for Fig 4a. I really had to zoom in on screen to see the α''_{lean} - which i think is just a continuation of α'' . Authors need to fix for readability. I'd like to see the colour scales in Fig 2 sorted out so the reader can see the minor phases more easily. Supplementary info should show some raw 2D detector images.

[AUTHORS REPLY 11]:

We thank the reviewer for pointing out these issues. We very much appreciate each suggestion that will make our figures clearer and easier to grasp. Therefore, we increased the font and symbol sizes in Figs. 3 and Fig. 4. In doing so we modified the vertical axis label of the plots in Fig. 3 showing the lattice parameter ratios to 'Edge ratio' to be more succinct. We also modified the color scaling in Fig. 2 so that low intensity reflections are better visible on the logarithmic scale. On the same line we increased the font and symbol sizes in Fig. 6 to improve the readability.

[General comments]:

We discovered a typographical mistake on Line 122 and wish to replace α''_{th} by α''_{iso} for the purpose of consistency.

To conform to the format requirements of Nature Communications following edits were carried out:

We slightly shortened the abstract to meet the 150 words limit maintaining its content. These changes are:

Line 18:

~~...diffusive versus displacive phase transformations permitting tailoring ...~~

Line 21:

~~We demonstrate that These ...~~

Line 23:

~~...two unique transformation pathways ...~~

Line 25:

~~Via the coupling of the ...both phases continuously evolve ...rejection of Nb.~~

Line 27:

~~...approaches not only for Ti-Nb alloys and for ...~~

Likewise we shortened a few formulation in the introduction, where a 1000 word limit applies, without altering its content or meaning:

Line 35:

~~...in modern aerospace design and in engineering...~~

Line 38:

~~Certain alloy compositions ...~~

Line 44:

~~...ever increasing interest in these materials.~~

Line 50:

~~The great diversity and the complex interplay...~~

Line 51:

...stimulate their further exploration ~~of these alloys~~.

Line 60:

It was reported that ~~the~~ decomposition ...

Line 68:

...during cooling ~~to~~ RT ...

Line 72:

...upon cooling after aging, ...

Line 74:

...diffraction methods permitting...

Line 81:

...serves as a prototype to study SM and SE...Its importance is underlined ...

Heating and expansion rates and were changed from /min and /°C to min⁻¹ and °C⁻¹, respectively.

Figure and Table titles were inserted into the captions:

Caption Fig. 1: Overview of heating-induced transformation behaviour.

Caption Fig. 2: Evolution of in-situ SXRD patterns. Results...

Caption Fig. 3: Heating-induced variation of unit cell geometries.

Caption Fig. 4: Martensite decomposition.

Caption Fig. 5: Close-up of the temperature interval of martensite decomposition for Ti-21Nb.

Caption Fig. 6: Formation of α''_{iso} .

Table 1: Thermal expansion rates of martensitic Ti-Nb alloys compared with the literature.

Table 2: Space group and Wyckoff position(s) for each phase used for the refinements.

Concluding we would like to thank the reviewers for their very constructive and detailed comments and suggestions.

Reviewers' Comments:

Reviewer #1:

None

Reviewer #3:

Remarks to the Author:

The authors have written a good and thorough response. I don't think they could be asked to do much more, and the manuscript is much more readable now. Its a complicated set of behaviours, so its always going to be slightly hard work for the reader.

Therefore I am happy to recommend publication.

Detailed Responses to Reviewers' Comments by Author

[REVIEWER'S COMMENT 1]:

Reviewer #1 (Remarks to the Author):

All my comments have been accounted for in the revised form of the manuscript.

[AUTHORS REPLY 1]:

We appreciate and thank the reviewer for his/her constructive remarks.

[REVIEWER'S COMMENT 2]:

Reviewer #3 (Remarks to the Author):

The authors have written a good and thorough response. I don't think they could be asked to do much more, and the manuscript is much more readable now. Its a complicated set of behaviours, so its always going to be slightly hard work for the reader.

On a personal note, I see that I have managed to be reviewer #3!

Therefore I am happy to recommend publication.

[AUTHORS REPLY 2]:

We are very thankful for the thorough and detailed review provided by Reviewer #3.